# Staff perspectives on the impact of COVID 19 on the delivery of specialist domestic abuse services in the UK: A qualitative study

Helen Riddell[1,2], Catherine Haighton[1] *

1 Faculty of Health and Life Sciences, Northumbria University, Newcastle upon Tyne, United Kingdom,
2 Public Health, Adult and Health Services, Durham County Council, Durham, United Kingdom

* katie.haighton@northumbria.ac.uk

## Abstract

Domestic abuse is a significant public health issue effecting 2.4 million adults in England and Wales each year. In March 2020 the World Health Organisation declared a global pandemic following the outbreak of COVID-19. As a result, the UK moved to a period of lockdown. There is growing evidence that highlights the unintended negative consequences of lockdown, particularly in households where abuse is present. The aim of this study was to explore the experiences of frontline specialist domestic abuse staff who continued to support victims during the period of lockdown to understand the impact of COVID-19 on service delivery. Ten, one to one, semi structured qualitative interviews were carried out with staff from a specialist domestic abuse service that operates in regions across the north-east of England. All participants had been involved in service delivery for a minimum of 12 months prior to March 2020 and had continued to deliver services throughout the UK initial lockdown period between March and July 2020. Each interview was transcribed verbatim, anonymised, then subjected to thematic analysis. Six themes were developed from the data covering: emergency support for victims; wider service efficiencies; victim safety; group work versus one-to-one support; criminal and family courts; and workforce development. While lockdown resulted in increased levels and severity of referrals, the switch to remote working brought a range of service efficiencies including time and money saved by negating the need to travel. Remote working also enhanced support offered to male victims and those with mental health issues but not those in rural locations with poor connectivity and those effected by the digital divide. Services should not underestimate the long-term benefits of peer support both to clients and staffs.

## Introduction

Domestic abuse is a significant public health issue, present within all social classes however there is recognition that low income, economic strain, and benefit receipt is associated with heightened vulnerability to domestic abuse [1]. It was estimated that 2.4 million adults (16–59 years) in England and Wales experienced domestic abuse in the last year [2]. National Police

**Data Availability Statement:** We have made excerpts of the transcripts relevant to the study available within the paper however there are ethical restrictions on sharing the de-identified data set.

The data contain potentially identifying and sensitive participant information and we do not have participant consent to share this dataset. Data requests may be sent to Northumbria University Research Ethics Committee (ref 24857) (nick. neave@northumbria.ac.uk).

**Funding:** The authors received no specific funding for this work.

**Competing interests:** We have read the journal's policy and the authors of this manuscript have the following competing interests: HR is lead commissioner for the organisation involved in the study.

data recorded 746,219 domestic abuse related crimes in the UK in year ending 31st March 2019, yet the majority of abuse goes unreported [3]. It is estimated that the current annual cost of domestic abuse in the UK is between £13.7 and £15.5 billion [4]. Domestic abuse is defined as "*any incident or patterns of controlling, coercive, threatening behaviour, violence or abuse between those aged 16 or over who are, or have been, intimate partners or family members regardless of gender or sexuality* [5].

In December 2019 a novel coronavirus (2019-nCov), a respiratory disease known more commonly as COVID-19, was identified as emerging from China [6]. It was quickly identified that COVID-19 could easily be passed between persons via respiratory droplets and through contact and fomites (an inanimate object that can carry a pathogen) [7]. The World Health Organisation (WHO) declared COVID-19 a pandemic on 11th March 2020 after rapid transmission throughout the world [8]. On 23rd March 2020, following a period if rapidly increasing transmission rates, the UK government introduced a range of measures which, in essence, placed the UK in what became commonly referred to as a nationwide 'lockdown' [9]. Since the declaration of COVID-19 as a global pandemic and the introduction of a range of restrictive measures by governments across the world in an attempt to slow transmission within the general population, there is an emerging evidence base highlighting multiple unintended and negative consequences; this was particularly evident in households and families where abuse was present [10].

According to the WHO there has been a 60% increase in emergency calls from women who are being subjected to intimate partner violence [11]. The WHO highlights the increase in calls across European Union (EU) states is directly correlated to countries being subjected to national lockdown measures [11]. In addition, reports are beginning to emerge that the restrictive measures that countries have been forced to adopt may have contributed to a future crisis of domestic abuse [12]. It is well documented that increased levels of additional stress or pressures on families where abuse may already be present can escalate the levels of abuse perpetrated. In the case of the COVID-19 global pandemic, despite the limited evidence base as to the true nature of the impact on victims of domestic abuse, some reasonable assumptions can be made [13].

Research that has explored the impact of other disaster scenarios has shown an increase in levels of family violence perpetrated [14]. Studies from Australia have cited increased levels of economic stress combined with the perceived inability of either perpetrator or victim to leave the family home due to COVID-19 restrictions as reasons for an increase in abusive behaviours whilst at the same time agencies have provided limited options for support [15]. In addition, research suggests that social isolation (a consequence of lockdown restrictions) may be a predictor of domestic abuse [16] and may be a factor in the forms of domestic abuse [17]. Extended periods in lockdown has led to already vulnerable people, experiencing heightened levels of abuse finding it more difficult to access support services [18]. Academics highlight that the prevalence of domestic abuse was at such high levels prior to COVID-19 that a further increase in prevalence may have long term, devastating impacts on society that may prove irreparable [12].

At the time of writing there were no studies of the impact of the COVID 19 pandemic on domestic abuse interventions, however a range of national charities and domestic abuse specialists have developed a range of guidance tools on how victims can be supported to be kept safe and how to continue to report abuse [12]. The aim of our study was therefore to understand the impact of COVID-19 on the ability of specialist domestic abuse services in the northeast of England to deliver support to victims between 23rd March 2020 and 6th July 2020 from the perspective of frontline staff to highlight innovative practice that services may wish to continue to develop and evaluate moving forward.

## Materials and methods

### Study design

We carried out one to one, semi structured interviews, with staff from a specialist domestic abuse service that operated across the north-east of England, adopting an interpretive framework. This framework guided the choice of method for data collection (interviews) and analysis (thematic). This design, based on phenomenology, helped us to understand the meaning of individual's lived experience [19]. We report this study and related findings in line with the Consolidated Criteria for Reporting Qualitative Research (COREQ) checklist [20] from the EQUATOR Network website.

### Ethical approval

We received ethical approval from Northumbria University Ethics Committee on the 8th August 2020 (ref 24857). Participants provided informed written consent to one of the authors (HR) prior to commencement of their interview.

### Sample

We used purposive sampling to identify frontline staff from a specialist domestic abuse service that delivered services across the north-east of England. Each participant was required to have been involved in direct service delivery for at least 12 months prior to March 2020 and responsible for the delivery of new interventions and adaptations throughout the specified dates between March and July 2020. This ensured that participants had good working knowledge of the topic area and were able to draw accurate comparisons between service delivery pre COVID-19 and during the national lockdown. Judgements about sample size, when to stop data collection and data saturation in thematic analysis are subjective, and therefore could not be determined (wholly) in advance of analysis [21] but based on previous research experience it was estimated that around ten interviews would provide sufficient data [22].

### Context

The voluntary sector organisation is well established and subsequently commissioned by several local authority areas across the north-east of England. The organisation delivers a range of specialist domestic abuse services for adult victims, perpetrators, children and young people effected by domestic abuse as well as refuge accommodation. These services include the Inspire Programme, an evidence-based female empowerment programme which aims to examine the attitudes and behaviours of perpetrators and the responses of victims and survivors. The Inspire Programme is delivered on a 6 week, rolling programme throughout the north-east of England. Due to the volume of referrals and the dispersed local geography there is a minimum of three programmes per week being delivered with the capacity for approximately 8–10 women on each course. Each local authority area develops their own service specification therefore service offers, delivery models and capacity differ considerably across the region. The organisation employs around 180 members of staff, equating to around 150 full time equivalents. A range of key interventions and existing processes were adapted, and national initiatives introduced, as a direct response to the national Covid 19 lockdown.

Services moved to a mainly remote delivery model, this included initial appointments, assessment work, safety planning and the delivery of the Inspire group empowerment programmes as well as updated referral mechanisms to prioritise those at most risk of significant harm. Wider partnership working also moved to virtual platforms, this included strategy meetings, Multi-Agency Risk Assessment Conferences, Child Protection Conferences and other

'Team around the Family' arrangements. In terms of staff support, all meetings and supervision sessions (group based and one to one), were delivered remotely and became less spontaneous due to the physical separation of staff and teams. Specialist counselling for staff members was also introduced.

From a national perspective, new interventions were introduced including the Rail to Refuge Scheme. The national government increased capacity in the National Domestic Abuse Helpline and other telephone/online support services. Adaptations were also made to the criminal justice service that included remote hearings of non-molestation orders. Direct behaviour change work with perpetrators was suspended meaning that in some circumstances elements of the domestic abuse system became more pressurised and services were limited to providing remote support to develop and/or review existing safety plans.

### Recruitment

As one of the authors (HR) is lead commissioner for the organisation involved in the study, she used existing working relationships to recruit participants onto the study. The voluntary sector organisation supported the recruitment process by identifying all members of staff who were eligible for the study and inviting them to express their interest in participating. HR emailed potential participants, who had expressed an interest in participating, to invite them to participate in the study. HR introduced herself as a female student and trained researcher (Masters in Public Health), rather than a commissioner, as she had existing work-based relationship with two of the potential participants and may have been known to other participants. HR provided potential participants with an information sheet and consent form and followed up each email with a telephone call. HR interviewed the first ten participants, from a list of sixteen, who agreed to take part in the study and returned the consent form. To support the interview process, we developed through discussion a topic guide which covered key areas of service delivery, including how clients made initial contact with the service, assessment, interventions, and exit (S1 Text). We piloted the guide throughout the first interviews and amended accordingly.

### Data collection

One of the authors (HR) conducted interviews between 28 August and 28 October 2020 using Microsoft Teams, due to the restrictions placed on face-to-face contact due to Covid-19, whilst audio recording each session using a digital recorder. At the start of each interview HR provided all participants with an overview of the study, this included the right to withdraw at any time and to reaffirm verbally their desire to participate. Interviews lasted up to 90 minutes.

### Data analysis

We adopted a thematic approach to data analysis [23]. Each one-to-one interview was transcribed verbatim, anonymised, then repeatedly reviewed to identify themes by one of the authors (HR). The themes identified were re-analysed so that they became more refined and relevant and were reviewed by both authors at regular monthly supervision meetings.

## Results

We present key findings in the form of the six main themes which were developed from the data along with verbatim quotations which best support these themes. Characteristics of the 10 participants are outlined in Table 1 below. All interviewees were female reflecting the characteristics of the organisation which had an all-female workforce. This is typical of the gender

**Table 1. Participant information.**

| Participant | Age Range | Gender | Length of Service |
|---|---|---|---|
| | (Years) | | (Years) |
| #1 | 35–39 | Female | 6–10 |
| #2 | 20–25 | Female | 1–5 |
| #3 | 20–25 | Female | 1–5 |
| #4 | 20–25 | Female | 1–5 |
| #5 | 26–29 | Female | 6–10 |
| #6 | 40–45 | Female | 10+ |
| #7 | 30–34 | Female | 10+ |
| #8 | 40–45 | Female | 10+ |
| #9 | 50–55 | Female | 1–5 |
| #10 | 50–55 | Female | 1–5 |

breakdown of staff in domestic abuse services and thus represents the work force. The age of the interviewees was categorised into five age groups ranging between 20 and 55 years.

## Theme 1: Emergency support for victims—*"I need to get out of the house, and I need to get out now"*

All participants reported increased referrals and higher caseload than before lockdown. Nearly all participants also reported the severity of incidents had increased, with three participants stating that they were dealing with increased recidivism:

> *". . .on the repeats, it's been like the ghosts of Christmas past I have had a lot of people come back in who I have dealt with over the years due to lockdown."*

(Participant 5)

Half of the participants reported that they had seen an increase in the level of physical assaults, often with alcohol as a contributing factor. Participants reflected that this was due to already volatile relationships under increased pressure due to lockdown restrictions using alcohol to 'cope'. This had then led to an escalation in abusive behaviours:

> *"I would definitely say that alcohol use was a big increase, I think people were bored so drinking more. I have people say well he had never hit me before; he was controlling and there was emotional abuse but suddenly over lockdown this has escalated and there's been a physical incident and usually alcohol is a contributing factor during lockdown."*

(Participant 2)

Two participants felt that they were seeing an increase in people contacting workers mid incident. One participant reported that calls, since lockdown, were more frantic. Both participants described occasions where clients had telephoned while trapped in a room with the perpetrator outside threatening them. On other occasions participants had received calls from clients saying they needed to leave the property immediately. This had impacted participants emotionally:

> *"I've seen an increase in things like "I need to get out of the house, and I need to get out now" or I'm locked in the bathroom and he's downstairs what can I do."*

(Participant 7)

Just over half of participants highlighted that there had been an increase in those seeking emergency accommodation, with a particular increase in requests from people who lived outside of the north-east of England. Three participants raised concerns that when they tried to contact other refuge providers, they reported that they were no longer accepting new clients due to COVID-19 which placed additional pressure on providers in the north-east of England who were continuing to operate throughout times of increased demand. One participant described having to prepare refuge rooms in under two hours to house victims quickly:

*". . .so, if I have my refuge hat on, we had to really look at what was coming in, in some cases we were turning rooms round in less than 2 hours it's been like crazy—a lot of out of area referrals"*

(Participant 7)

Half of the participants identified the newly introduced Rail to Refuge Scheme as an invaluable addition to the national domestic abuse system supporting those fleeing abuse. They described the scheme, which allows victims of domestic abuse to travel (free of charge) on the UK rail network to be able to access emergency accommodation anywhere in the country, as one of the most positive initiatives that had been developed nationally over this time:

*"The rail to refuge scheme was really good too, that worked really well, we had at least two people that I know of arrive here using it and I know one person who travelled down south using it but I think there were others, I hope it stays."*

(Participant 4)

## Theme 2: Wider service efficiencies—*"There's been a lot less no shows and I'm saving so much time"*

All participants identified issues with non-attendance of clients at prearranged face-to-face appointments and unsuccessful home visits as a significant issue prior to the switch to remote delivery. This was coupled with the difficulties of working across a large geographical area and the negative impact that travelling had on their working day. One participant described how she would usually arrange three face-to-face appointments per day and of these it was common for at least one or two to be unsuccessful. All participants reported that they felt they wasted time travelling and were disheartened when clients would not attend:

*"Before lockdown I would probably have one out of about three turn up most days, it would be a very rare day if three out of three would attend and sometimes that would mean going up and down the county for nothing."*

(Participant 2)

Since remote working had been introduced all participants reported that they were able to contact a higher number of clients per day and were more productive as they had a much higher rate of clients attending their appointments. It was felt, by all the participants, that clients were more likely to answer their phone than turn up for a face-to-face appointment:

*"The fact [is] we have more time. We all cover a huge area and now I work in [the] office two days per week. On office days I can manage two or three face-to-face appointments whereas*

*when I am working remotely, I am seeing up to six or seven per working day, but if I'm out and about referrals are just sat there."*

(Participant 1)

Most participants also highlighted that they had observed better engagement from clients who had mental health and/or anxiety issues. Participants reflected that these clients would often be the people who would cancel appointments at the last minute or not show up at all and that now they had the opportunity to connect over the telephone participants had seen much quicker initial engagement:

*"Actually, I'd say there has been a lot less no shows by phone. A lot of people have mental health and anxiety issues and struggle to leave the house, in the community venues we got a high level of no shows, but this has definitely been better for them."*

(Participant 4)

Just over half of participants talked about how they felt clients were more open since carrying out assessments over the telephone. These participants described feeling that clients were speaking much more freely and that they were developing good rapport despite it being a telephone call. One participant felt that clients were disclosing incidents of sexual or familial abuse much earlier in the process than usual and felt this was solely down to the anonymity of being able to be heard but not seen:

*"People seem to be disclosing a lot more, more quickly. It would more likely take about three to four sessions for someone to disclose sexual abuse face-to-face or familial abuse, now they are telling us that it's happening in the second session."*

(Participant 8)

Just over half of participants also discussed the benefits of telephone over face-to-face contact to themselves and the clients because they were able to be more flexible to meet both of their needs. All participants gave examples of how they could rearrange appointments much more easily and could make telephone calls outside of office hours; this was particularly helpful for clients and participants with children:

*"...we have the flexibility to meet the needs of the client in terms of appointments and things too, now we can do evenings or weekends or whatever fits best."*

(Participant 8)

A few of the younger participants stated that previously their age would be an initial barrier to some clients engaging with them. They stated that clients would often ask how someone who looked so young could possibly have enough experience to help and support them. For these staff members remote working had taken away that pre-existing barrier, and they reported that this made them able to focus on the work with the client without the need to prove themselves capable:

*"This might sound strange but my thing that's improved is around age, so because I'm twenty-five anyone that's 30 or younger, if I do a face-to-face with them no issues what so ever, but*

*sometimes I can sense that when the people are 40–50 they are thinking how is this person going to help me."*

(Participant 2)

All participants reported that they felt more productive working from home, and one participant referenced an improvement in her work life balance as she was cutting out travel time and was able to spend more time with her family. Some of the participants felt that their record keeping had improved as they described how they were now able to type into documents whilst speaking to clients and update the case management system immediately after a contact. Previously they would make notes during appointments and have to return to the office to type them up and update systems:

*"For me I think I'm getting better, I'm a lot more productive, I'm able to take a call and type while I'm on the call but when I'm in a client's house I make notes but still need to type it all up and there's the travel on top."*

(Participant 3)

Despite the positive reflections about increased levels of engagement, all participants reported some challenges in operating in a remote service such as what they potentially missed by not having face-to-face appointments:

"*The only thing is I could be doing an assessment with some girl over the phone and she could be sat there with a broken arm and 2 black eyes and I don't know that do I and there might be no police report or nothing so when you look at it like that the nature of what we do there could be people slipping through the net because verbally over the phone if they don't want to tell you they don't have to and I'm not seeing anything else.*"

(Participant 9)

Almost all participants talked about the importance of body language and non-verbal cues as indicators within direct face-to-face work. Participants described how, when completing assessments for example, participants reported knowing when to probe further based on how a person physically reacted to a question just as much as from what they said. They also felt that this helped participants determine whether they thought a client was able to disclose the extent of their abuse:

*"In person you can judge the body language and decipher if someone is being truthful or not. Some people you just get yes or no's but in person you would be able to wangle more out of them, so it has positives and negatives really like everything."*

(Participant 4)

Half the participants stated that the shift from face-to-face to telephone contact created some unintended inequalities which related specifically to the rurality of some of the villages and the lack of mobile phone signal and internet connection. Of these, a further two participants raised concerns about the ability for some clients to access their service due to a lack of equipment. This included examples where the perpetrator had purposely damaged mobile phones and there was a lack of access to resources to replace them.

*"Yeh, signal is a nightmare for some as we have lots of little villages and places where signal is terrible and that has made some things really difficult. Some people just haven't had the technology and can't afford it."*

(Participant 3)

While acknowledging the clear positives expressed, some participants also highlighted several issues with working from home arrangements. Half the participants reported that they missed the support of their colleagues in the office environment, that the ability to informally talk over scenarios or ask a quick question had been lost and that from a mental health perspective the arrangements had not always been positive for them:

*". . .a lot of our role in keeping your own mental health ok is the ability to off load on each other informally too and that's missing, so we need to work hard on making opportunities and looking after people. I mean people are sat at home all day discussing these really hard going and hard-hitting cases"*

(Participant 7)

A couple of participants highlighted the challenges working from home had placed on either themselves or colleagues who have children and how managing to safely work and look after children had placed significant extra pressure on households. One participant highlighted that she could be easily distracted at home and that at times this had been very challenging to manage, she described that when she was in a client's home, she was completely focused on them and their surroundings and that she had had to work hard to ensure that she stayed completely focused when now delivering remotely:

*"My concern is for myself; it's about making sure I'm fully focused when I'm on a call not being distracted by emails that are popping in because I'm recording directly onto a word document. When I'm in their home my full attention is on them at home there's these other distractions."*

(Participants 2)

### Theme 3: Victim safety—*"Is it safe to talk?"*

Safety was a key feature through all interviews. Every participant made multiple comments on differing aspects of safety, including contacting clients, assessments, safety planning and delivering interventions.

All participants described that prior to lockdown, staff would always meet clients face-to-face at an agreed time and venue, either in their home or in a suitable community location. This way they could be sure who was in the family home, check conditions and guarantee they were alone. However, once all contact and support transferred to telephone contact this became more challenging.

Almost all the participants highlighted that they made sure that a code word was in place with every client even those who reported that they were no longer in the relationship; clarified at the beginning of every contact alongside checking if the client was in a safe space and if they could talk freely. The remaining participant worked predominately with children and young people which made some of these checks unnecessary, however they also described asking young people that they were able to talk freely.

*"I would always start by asking if it's safe to talk, the referral usually tells us who is still in the home so we have an idea then I would use the code word if there's one set up or I would set a code word up with them. We would always ask if they were in a safe space and could talk freely."*

(Participant 3)

Most participants identified positive changes to Initial Assessments and Support Plans because of lockdown. Participants outlined that the new *COVID-19 Secure* assessment and plans included additional safety questions that helped establish who was in the property and at what times, providing greater understanding of safe and danger times. Much greater focus had been put into understanding the layout of each client's property and, as part of a safety plan, emphasising with the client what were danger rooms, for example, kitchens as they have a greater number of items that can be used as a weapon and how to try and keep away from these areas should an incident start to develop. Some participants described the importance of checking that the client contact information was keep up to date and checked at each contact. Two participants described how they talked with the client, in depth, about how technology can be used as a method of control and therefore needed to be considered as part of a safety plan.

*"We also have a lot more emphasis on technology and phone safety and checking on their support network—including things like video calls, shared devices as these are used to control too"*

(Participant 3)

However, despite safety checks, participants raised concerns for those clients who were still in a relationship with the abusive partner. Most of the participants reported that, as part of any assessment, they tried to identify what opportunities there may be to talk with the client alone but recognised for some this was not possible therefore felt like some clients were not able to receive the full support they needed.

*"I would say specifically one client stands out, she did say she wanted to engage but was in the relationship and wanted to continue the relationship but she just couldn't engage because of the circumstances, she didn't respond to phone calls but she would send random emails but I would reply and offer a time for a call but whenever I called her mobile would always be switched off—I did that for about a month but got nowhere"*

(Participant 1)

Many participants raised concerns that people were not always truthful with them, often hearing other people in the background, this was mentioned specifically in relation to clients referred by Children's Social Care. Participants felt that these clients were often forced by social workers to engage with the service and therefore contact was superficial. Participants reported that these scenarios made them worry as to whether they should carry on but in most cases they felt they needed to trust the clients were being honest.

*"The main thing is establishing if they are safe, they can tell you they are, but you can't physically see them you just have to take their word for it. You just don't know if you have the full picture"*

(Participant 4)

### Theme 4: Group work versus one-to-one support *"The biggest change has been to groups . . ."*

It was quickly identified throughout each interview that the change to service delivery, which had the biggest positive impact from a client and service perspective, related to group work support moving from face-to-face sessions to delivery remotely over the telephone on a one-to-one basis. It was reported by all participants that these changes, particularly to the Inspire Programme have had the biggest impact on them as workers and the clients referred for support.

Nearly all participants reported that the ability to deliver the Inspire Programme on a one-to-one basis had significantly improved their practice; the only participant who did not provide a supporting statement about this element was not currently involved in the delivery of adult focused services. Of those participants who did deliver adult interventions, all but one reported that, despite feeling anxious before delivery, they now felt their practice had significantly improved and that the programmes associated resources had proved a wider valuable resource to apply to the practice beyond the scope of the Inspire Programme. The remaining participant previously delivered Inspire and whilst had no anxiety about one-to-one delivery acknowledged that she had also developed new skills.

All but one of these participants highlighted that, because the programme was now delivered on a one-to-one basis, for the first time they felt they had something more substantial to offer male victims of abuse. Previously these groups were made available for female clients only and their content was not suitable for male victims. Participants strongly expressed that this has had a major impact on enhancing services and the support they were able to offer:

> *"One of the biggest positives is that on a one-to-one basis you can deliver it to male victims which is brilliant because we have never had anything group wise or longer intervention wise for men to engage in. That's been a massive help, we have changed pronouns and been able to focus the content for that individual."*

(Participant 2)

Over half of the participants stressed that the remote, one-to-one style of delivery meant that clients who they felt would never have attended groups, specifically those with mental health and/or anxiety issues could now benefit from a much more intensive and meaningful support beyond safety planning and more general outreach. It was also reported that, because childcare was not provided for face-to-face group support, a high number of clients who would really benefit from group work did not and previously there would be no alternative offered; but with the new one-to-one way they reported much higher retention:

> *". . .we always say if you have three kids, no childcare and have to get a bus there's no way they are coming to a group."*

(Participant 8)

Over half of the participants highlighted that previously all clients referred for group work would be passed to a different member of staff within the organisation, however they felt that by embedding the content into the existing support plans it was much better for the client as it involved less hand overs between staff, client's no longer needed to repeat any parts of their story and that it helped develop more meaningful relationships and trust between the client and the member of staff providing all elements of the support package. All these participants

stated how beneficial it had been for them to see the direct impact of the support they had delivered where previously this would have been a referral to someone else:

> *"What's also good is you have been on a journey with them now, you did assessment, you have done outreach and now your delivering group, we are much more trauma informed."*

(Participant 2)

Most participants stated that the ability to deliver support on a one-to-one basis meant that they were able to tailor the sessions to the needs of the individual whereas in the group setting this would not be possible. Half of the participants highlighted that for those young people aged 11 years and over the switch to one-to-one telephone support had seen a significant increase in engagement. Previously children and young people would be offered attendance at either an under 11 years group or over 11 years group. Participants described how the large range in age was often off putting for the older children and that the changes in delivery had made a significant difference.

> *". . .so, for me putting three-year olds in with eleven-year-olds just doesn't work, personally I just don't feel like we are supporting them enough because you can't do one activity that meets the needs of everyone. We did have parents say that the child won't attend because they were in a group with babies"*

(Participant 6)

Whilst the interviews generated a significant level of positive statements in relation to the changes to the group work model, most participants also acknowledged that by only delivering on a one-to-one basis, clients missed out on the positive aspects of being part of a group. All participants identified that a key feature of group work is the benefits of gaining peer support and feeling less isolated as they heard others share their similar stories and experiences. Interview participants outlined that those clients who did participate in group activities reported feeling more confident and often developed long-term friendships with other attendees. Two interview participants highlighted that those clients who attended group programmes together often moved through the service and into the wider recovery element together and without that in place they had seen a drop in people transferring into recovery:

> *"The thing that I think's missing is that when you're in a group and you talk about the power and control wheel or the tactics that perpetrators use, people sit and they nod and then everyone in the room is agreeing and also seeing that they aren't alone, they weren't going crazy. I have had feedback where people have said that they feel so accepted because for years he made me feel alone but here everyone is nodding and I know I'm not going crazy. It's just the peer support is phenomenal"*

(Participant 2)

Half of the participants expressed concern for the lack of group work provision for children under the age of 11 years. While these children had not been able to access any direct support remotely, workers had been providing parents with some tools and activities to carry out with their child as an alternative. Participants were concerned that these younger children may continue to be at risk or that they were not having a positive experience:

*". . .the younger ones sending sheets over to parents you do wonder whether it's doing the right thing, I mean are they delivering it properly, is that child getting a positive experience but I mean it was the best we could do because with the best will in the world you can't do a phone call or Whatsapp with a five or six year old, it just wouldn't be safe. I found that quite frustrating because I couldn't help everybody, I mean I had that weekly contact and if I could help I would but it's just so hard and like how do you get round that."*

(Participant 6)

Participants also raised capacity issues since one-to-one group work had been introduced, half of the respondents highlighted issues with large and increasing caseloads to manage leaving them feeling overwhelmed.

*"I think from a personal note—it's been really stressful, it's been really high paced, and caseloads are massive, and it's just felt like they were never going to go down. . .sometimes I feel overwhelmed"*

(Participant 2)

## Theme 5: Criminal and family courts—*"the courts and connected processes basically just stopped"*

Just over half of participants raised the issue that courts, which already struggled to meet the demand prior to lockdown, 'just stopped'. Participant's concerns centred around the impact further delays had on victims, they described that perpetrators, most common in the family courts where child contact is being arranged, often use these processes to further control. The delay has meant that these scenarios are ongoing without an end date and can place further stress and upset in households.

*"there were a few women who were on the groups and they were waiting for the final hearing in family court to get their kids back so there was a lot of upset and they were messaging me to see if I could do anything because they couldn't go to court and it meant they just couldn't get their kids back it was heart-breaking for these women to be so close and worked so hard and done everything asked of them despite what the perpetrator was doing in some cases then just to have no idea when it was going to happen was devastating"*

(Participant 10)

Participants went on to describe the impact on criminal courts as one of the most challenging aspects of lockdown. Perpetrators, often released on bail, but waiting for court dates placed significant additional worry on the victim because they knew they were still out in the community:

*"I have cases from last November that are still waiting to go to court, imagine how the victim feels about this. I mean perps can be on bail yes, they might have bail restrictions, but they are out and about in the community and the victim's is still terrified."*

(Participant 5)

Despite these concerns, participants were able to highlight some positives on court processes, with half of the participants highlighting the improvements that have been

implemented to the granting of Non-molestation Orders. They described that prior to lock-down, a victim was required to attend court with a solicitor and request that a judge grant the order by providing evidence of its need, a stressful and intimidating process for victims. However, since lockdown these orders had been granted remotely. Solicitors have been able to facilitate three-way telephone calls that allowed the victims to speak directly with judges to have the orders granted. Not only was this less stressful for the victim, the participants' reported that it had significantly reduced waiting times.

> *"I have one client I spoke to last week, I put the referral in for civil support then a few days later she called me and said thanks so much I have my non molestation order, had a really nice conversation with judge and solicitor in chambers over the telephone and it was 15 mins and I didn't have to say much. She was over the moon."*

(Participant 1)

## Theme 6: Workforce development—*"We learnt how to use Teams in a couple of days. . .."*

The final theme raised by participants was the adaptations to practice that workers were required to make, and the accompanying workforce development package and additional emotional health and wellbeing support made available. Whilst most participants highlighted that they had received some dedicated online training, the feedback on the quality was variable although it was acknowledged that the sessions were put together very quickly under pressurised circumstances.

Half of the participants felt that the use of Microsoft Teams accelerated very quickly as it became the main contact method for key partner meetings, particularly Children's Social Care and Early Help services. Some participants felt that their supervision sessions were more productive via Teams as they were able to have face-to-face contact with their manager without the need to travel long distances to office bases.

Some participants raised concerns for new starters to the organisation, reflecting the difficulties for someone new to join under remote working circumstances. Participants identified that they were missing out on opportunities to shadow experienced staff and ask questions in an office environment as they felt this was how you became embedded into a team and how you learn most effectively:

> *"You learn from hearing others too, like I said for new people it's really hard, you learn more from others than anything else when you're new. It must be so hard for them, there [are] no shadowing opportunities or relationship building to help develop our own style"*

(Participant 1)

Some participants highlighted the introduction of specialist counselling for staff members to access as a real positive for the service. This was counselling that had been made available to staff specifically to support their emotional wellbeing and mental health whilst solely remote working and was introduced to support staff following the death of a former colleague due to COVID-19:

> *". . .staff counselling for us was introduced which is now in place and people are really utilising that to help them through and lots of the team access that service."*

(Participant 6)

## Discussion

In this UK-based study of domestic abuse services delivered during the COVID 19 pandemic, we found that while lockdown resulted in increased levels and severity of referrals the switch to remote working clearly brought a range of positives to the worker, the client, and the service.

International literature confirms our finding of a marked increase in the level and severity of domestic abuse incidents [15] and has also highlighted our finding in relation to the increased use of alcohol as an emerging feature reportedly linked to stressful events and lack of external support networks [14]. However, as we found in our study there is also evidence for the effectiveness of technology in delivering healthcare services [24, 25], particularly in rural areas [26], supporting remote working. Over the last ten years specialist telephone mental health assessments have been deemed appropriate and relatively common place, especially with older populations [27]. This is in line with more recent studies relating to the success of broader mental health interventions delivered through platforms including smartphones [28].

Evidence also corroborates our finding that suggests that survivors of domestic abuse are open to and in some cases prefer technology-based interventions [29]. However, as we have found, remote working requires an enhanced level of safety measures in place to ensure that clients are able to access support services safely and without negative repercussions [29]. Our study has however also highlighted the need to be more aware of the ways technology can be utilised as a method of control a finding replicated in other research [30].

Despite our study providing evidence for the effectiveness of remote working, those who prefer face-to-face contact, or who do not have the resources to engage via remote methods, may be at a disadvantage. Issues around digital inclusion across the UK have been well documented over recent years and it is recognised that not all populations are equally served [31]. In addition, children and young people from more disadvantaged backgrounds may also face additional inequalities from the 'Digital Divide' [32] yet are at increased risk of long-term disadvantage if they are exposed to domestic abuse [33]. This, alongside the rural location, means that any future service model must take into consideration that nationally only 41% of rural homes have 4G coverage and in some areas no cover at all [34].

From a client perspective, our study showed that remote working has its strengths, in particular supporting more vulnerable clients for example those with mental health and/or anxiety issues. However, this should not come at the detriment to other clients for whom remote working is either unsafe or for those who face additional digital inequalities in terms of access to appropriate equipment and/or live in an area with limited connectivity. Client safety must be at the heart of all service development, with this in mind support services need to ensure that they are developing tools and resources that allow workers to make sound judgements when working remotely. When thinking about safety, services must also consider that when operating remotely the lack of ability to interpret non-verbal cues and body language may impact on the long-term effectiveness of an assessment and/or intervention.

As highlighted in our study, the ability of staff to interpret non-verbal cues and understand when to probe more is fundamental in determining the validity of the client's response [35] much of which has been lost due to remote working. In addition, a recent study exploring the impact of remote working on wellbeing during COVID-19 has highlighted that the additional pressures of balancing home life with increased workloads in a remote setting without positive organisational cultures can have a negative impact on employees' wellbeing [36]. In our study remote working appeared to be a cost-effective delivery model saving valuable time and

resources. However, this must be properly resourced at an organisational level to ensure that staff have access to quality resources, the appropriate level of support in place including opportunities to stay connected to their peers to maintain a healthy work—life balance and positive emotional wellbeing. Investment is required in more bespoke offers of induction packages and longer-term workforce development programmes. This is particularly relevant for newly appointed employees to build connections with wider team members whilst operating a heavily remote led service model.

Domestic abuse is widely accepted as a gendered issue with feminist theories and analysis of men's violence against women adopted by some of the world's most prominent institutions [30] and thus dominated the development of support services across the UK. However, the view that men can also be victims is less accepted or researched [37] and there are limited studies describing support and interventions for male victims [38]. The results of this study highlighted that although relatively low numbers of male victim's request support locally, when they do reach out the offer of support is limited. Often men perceive that they hold no place within domestic abuse support services or that the interventions are not appropriate [38], the lack of support offered previously may reinforce this perception. Our study revealed that remote working enhanced support offered to male victims. The ability of the study participants to transform and adapt previously female orientated resources and interventions with reported successful outcomes, highlights the broader need for services and policy to recognise what some academics have termed a gender aware domestic abuse culture that encompasses the needs of male and female victims [38]. Whilst domestic abuse services remain a space for female only victims, it may further hinder male victims of abuse seeking support [39]. The area of men as a victim of abuse and the support they receive is an area that would benefit from future study.

Peer support developed through group work learning was not available during lockdown. Previous studies have highlighted that women in peer group recovery services receive essential messages of inclusion and belonging, enhanced empowerment and move from victim to survivor, to thriving by participation in group empowerment programmes [40]. Similarly to adults, studies have highlighted the positive effects that peer relationships can have in developing protective factors for children and young people who experience domestic abuse [41]. Services should not underestimate the impact of empowerment groups and peer support on clients moving forward.

## Limitations

One of the authors (HR) had an existing working relationship with the participating organisation and this may have influenced the interviewees responses resulting in socially desirable responding or the tendency of respondents to reply in a manner that would be viewed favourably by others. Participants were all female, which reflects the characteristics of the organisation which has an all-female workforce. In addition, a predominantly female workforce is typical of domestic abuse services and so the sample was broadly representative. The study explored the views of specialist domestic abuse workers from one organisation covering a single geographic area therefore it may not represent the views of all services operating at this time. However, since domestic abuse services in many countries across the world have faced similar challenges, our findings are likely to be relevant beyond this specific setting and similar research in other contexts is warranted. It should also be noted that the study had a relatively small sample therefore potential issues regarding data saturation may have occurred. In addition, face-to-face interviews with participants may have improved or varied responses. As far

as we are aware however this is the first study to explore the impact of COVID-19 on specialist domestic abuse services from a worker's perspective.

## Conclusion

In this UK-based study of domestic abuse services delivered during the COVID 19 pandemic, we found clear evidence that can be applied to the development of future specialist service delivery models that may support efficiencies and promote positive relationships. We would suggest retaining some elements of secure remote delivery particularly for clients who prefer this type of service or those with mental health issues. Home visits are still important however especially for those who are digitally disadvantaged and for young children. We would also suggest that empowerment programmes such as the Inspire Programme are offered both on a one to one/remote basis (particularly for men, women with mental health or childcare issues, and older children) and on a group basis for women and young children who would benefit from peer support. We also advocate the ongoing use of peer support and specialist counselling for (particularly new) staff. We would also encourage the retention or implementation of similar national schemes to the Rail to Refuge Scheme as it removed pressure on both victims and services by allowing victims of domestic abuse to travel free of charge to be able to access emergency accommodation anywhere in the country. Retaining remote hearings for granting of injunctions such as Non-molestation Orders would reduce stress and wait time for victims. However, these suggestions should be treated and applied with caution to prevent the development of further inequalities.

## Supporting information

**S1 Text. Interview topic guide.**
(DOCX)

## Acknowledgments

We would like to thank all of the participants who gave up their time in order to contribute to this study.

## Author Contributions

**Conceptualization:** Helen Riddell, Catherine Haighton.

**Data curation:** Helen Riddell, Catherine Haighton.

**Formal analysis:** Helen Riddell, Catherine Haighton.

**Supervision:** Catherine Haighton.

**Writing – original draft:** Helen Riddell, Catherine Haighton.

**Writing – review & editing:** Helen Riddell, Catherine Haighton.

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
