## [Decision Letter · Decision Letter 0]

18 Jan 2022

PGPH-D-21-00908

Staff perspectives on the impact of COVID 19 on the delivery of specialist domestic abuse services in the UK: A qualitative study

Dear Dr. Haighton,

Thank you for submitting your manuscript to PLOS Global Public Health. After careful consideration, we feel that it has merit but does not fully meet PLOS Global Public Health’s publication criteria as it currently stands. Therefore, we invite you to submit a revised version of the manuscript that addresses the points raised during the review process.

Reviewer 1 has made a suggestion to condense the manuscript. Please review the comment and see if there are areas you would like to condense. However, PLOS Global Health does not have a word limit, and so you are not required to reduce the text significantly if you do not feel you can do so while still adequately conveying the necessary components of the manuscript.Reviewer 2 has suggested that "Every paragraph in discussion should be linked to something you found and want to discuss relative to external literature". Please consider how you can better tie in your discussion to the existing literature, but not every paragraph needs to do so as suggested.

We look forward to receiving your revised manuscript.

Kind regards,

Tia M. Palermo

Academic Editor

Journal Requirements:

1. Please update the completed 'Competing Interests' statement, including any COIs declared by your co-authors. If you have no competing interests to declare, please state "The authors have declared that no competing interests exist". Otherwise please declare all competing interests beginning with the statement "I have read the journal's policy and the authors of this manuscript have the following competing interests:"

2. We notice that your supplementary information is included in the manuscript file.  Please remove them and upload them  with the file type 'Supporting Information'  . Please ensure that all Supporting Information files are included correctly and that each one has a legend listed in the manuscript after the references list. 

3. We have noticed that you have uploaded supporting information but you have not included a list of legends.  Please add a full list of legends for all supporting information files (including figures, table and data files) after the references list. 

4. Please amend your Data Availability Statement and indicate where the data may be found.

Additional Editor Comments (if provided):

The reviewers found value in this manuscript on the topic of domestic violence service provision in the context of COVID-19 and have made suggestions for how to improve it, including revisions to the discussion and a description of a theoretical framework tied to the methods. There are also some suggestions on how to improve clarity in the methods.

Also please note that PLOS Global Public Health, for qualitative studies, typically asks that researchers make de-identified data available, along with the data collection forms they used. Other exceptions are detailed in our policies, below:

For studies analyzing data collected as part of qualitative research, authors should make excerpts of the transcripts relevant to the study available in an appropriate data repository, within the paper, or upon request if they cannot be shared publicly. If even sharing excerpts would violate the agreement to which the participants consented, authors should explain this restriction and what data they are able to share in their Data Availability Statement.

See the Qualitative Data Repository for more information about managing and depositing qualitative data. See policy: https://journals.plos.org/globalpublichealth/s/data-availability

Our team will work with you at acceptance (if it is accepted) to ensure the data availability statement is clear and contains the necessary information.

Reviewers' comments:

Reviewer's Responses to Questions

**Comments to the Author**

1. Does this manuscript meet PLOS Global Public Health’s publication criteria? Is the manuscript technically sound, and do the data support the conclusions? The manuscript must describe methodologically and ethically rigorous research with conclusions that are appropriately drawn based on the data presented.

Reviewer #1: Yes

Reviewer #2: Yes

2. Has the statistical analysis been performed appropriately and rigorously?

Reviewer #1: N/A

Reviewer #2: N/A

3. Have the authors made all data underlying the findings in their manuscript fully available (please refer to the Data Availability Statement at the start of the manuscript PDF file)?

Reviewer #1: No

Reviewer #2: No

4. Is the manuscript presented in an intelligible fashion and written in standard English?

Reviewer #1: Yes

Reviewer #2: Yes

5. Review Comments to the Author

Reviewer #1: Thank you for the opportunity to read and review this paper, on the topic of UK domestic abuse services in the context of the covid-19 pandemic. This is an issue of obvious public health relevance, and this study can provide a relevant contribution. Provided that some revisions are made, as pointed to below, I would recommend this study for publication.

1 I think the discussion should delve a little bit more into what insights can be generalized and carried into a ‘back-to-normal’ versus ‘staying-online’ state of affairs. In the limitations section, the authors do state that this study is made in a particular context and that the results may therefore not be representative for other services/places, but I think this issue should be discussed a little bit more since domestic abuse services in many countries across the world have faced similar challenges. If nothing else, the authors could make a claim that findings and themes are likely relevant beyond the specific setting, and that similar research in other contexts is warranted. Also, while we don’t know at this point when and to what extent services will go ‘back to normal’, the authors could discuss a bit more clearly which adapted practices should be kept in place or developed and which should revert back to face-to-face praxis, if the latter is possible, or what should be kept in mind should services need to be kept, or temporarily return to, online-only.

2 The manuscript could be condensed. At the moment it’s quite long, and while it is helpful to get this much detail and this many quotes, I think the readability would improve if some quotes (which don’t add substantially to what is already mentioned in the text) were omitted and the text revised with abbreviation in mind. I think Table 1 is superfluous, as it doesn’t provide information that isn’t or can’t be stated in the text (i.e., the age of the participants was between 20 and 55 years, and half had 1-4 yrs of experience while the other half had 5 yrs or more).

Along related lines, the manuscript would benefit from being proof-read throughout. Many sentences are very long (e.g., line 274-278, 299-304), some are a bit unclear (e.g., p 4 “Learning from studies… has shown”; p 10 “Participants reflected that this was due to the nature…” and “Both participants described calls…”; p 167 “All participants commented on…”) and others are not complete (e.g., sentence starting on line 99). The paragraph starting on line 64 is long, and should be broken up into two. While there are different approaches to quotation, I would recommend slightly editing some of the quotes for punctuation and syntax (e.g, line 244-247, 269-272).

3. Regarding the decision on the number of interviews, I think the reference “based on previous work” is a bit vague. This is perhaps particularly so as the authors refer to piloting throughout the first interviews and making amendments accordingly. Did the authors find that they reached a level of saturation in the material? Also, with regards to sampling via author e-mails, the authors state that their “approach was used as there were existing work-based relationship with two of the potential participants and the author (HR) may have been known to other participants”. This is a bit unclear to me. This method was chosen above what alternative?

4. I think it would be helpful if the authors elaborate a little bit, before or in the beginning of the results section, about the changes in practices made due to the lockdown, including the special measures that were introduced. Now such measures are mentioned here and there the results (Rail to Refuge, COVID-19 Secure, remote granting of non-molestation orders, workforce development package/additional emotional health and wellbeing support). While this is all good, I think an overview of changes and adaptations – including ones that perhaps aren’t mentioned in the paper at this point - would be useful.

5.

- On p 6, it would be interesting to find out early on (although it is mentioned later that it is voluntary) if the service provider was voluntary or part of a state-funded social service. Some more information about it may also be interesting: how many people it employs/involves, how many clients it serves... Also, please provide a reference to the COREQ checklist.

- On p 8, regarding the analysis, the authors state that “The themes identified were re-analysed so that they became more refined and relevant and were reviewed by both authors on an ongoing basis”. Could this be a specified slightly – what does “on an ongoing basis” mean?

- On p 2, “connections with socio-economic disadvantage and social welfare” is a bit vague - can this be specified somewhat?

- A few times, the word “business” is used for the service provider activity. Is this an appropriate concept to use?

- On line 205, does “make ready refuge rooms” mean prepare rooms or make new ones?

- Is “recidivism” a better term to use than “repeat”?

- On p 3, “ending March” – does this mean ending in March, or ending on March 1st/31st?

- On p 5, the very specific time frame doesn’t seem to rhyme with the qualitative approach used in the paper, unless there is a particular reason for it.

- Finally, in the introduction the authors list a few aspects of how lockdown may affect domestic abuse. A suggestion, although this is up to the authors, is that the relevance of social isolation as a correlate to [1] and as central to the dynamic of [e.g., 2] (forms of) domestic abuse is also mentioned.

[1] Lanier C, Maume MO. Intimate Partner Violence and Social Isolation Across the Rural/Urban Divide. Violence Against Women. 2009;15(11):1311-1330. doi:10.1177/1077801209346711

Social isolation correlate –

[2] Leone, J. M., Johnson, M. P., & Cohan, C. L. (2007). Victim help seeking: Differences between intimate terrorism and situational couple violence. Family relations, 56(5), 427-439.

Reviewer #2: This paper provides important information about the context of DV service provision in the Covid-19 pandemic and can be useful for other organizations that are similarly adapting their work. The paper would benefit from some edits, as noted below, and in particular the discussion needs to be rewritten and description of a theoretical framework or foundation should be tied into the methods section. While generally well-written, copyediting is needed to rectify some commas in the wrong place throughout the paper and to remedy overuse of the passive voice, especially in the methods section. I look forward to seeing a future version of this important paper, which I hope will be published after appropriate revisions and will benefit the field.

The lack of data availability is suitable for the field of domestic violence, though it would be useful to state that "data are available upon reasonable request".

Abstract

• A growing number of qualitative researchers, notabely Braun and Clarke who wrote one of the most cited guides to thematic analysis, and who you cite (citation 17)

argue that themes can’t “emerge” – suggest you update this language (in abstract and results section).

Introduction

• You talk about the 24% increase from 2018 to 2019 in violence – before the pandemic. Why is that increase relevant, and do you have ideas to share or evidence on why that is the case? Could the further increase once Covid began not be in part a continuation of that trend? Not sure why the 24% increase figure is relevant to your narrative.

• page 4 row 74- “Learning from” is not needed - start with “Studies show”

Methods

• You talk about this as an evaluation study (filling the gap in terms of no evaluations) but the study design does not mentioned evaluation or what type of evaluation design you use.

• p 6 study design – not a full sentence- no verb and also too long of a sentence. Be direct “We carried out” .

• Could you delete “firmly within the boundaries of” - Not sure what that means in the context of the sentence. Also you don’t cite or define phenomenology

• SAMPLE: in March and July 2020 (did you mean “between”? What about April, may, June)

• It is unclear what you mean by p 7 row 131 “This approach was used…” – please clarify what you mean by this sentence- what approach, why did that help the situation. what does it mean to be known to other participants and how does that influence things and your approach. Be specific.

• Interview guide development – how did you come up with the 6 areas? Does it draw on literature – if so cite. Or on past studies? This relates to the next point.

• Not theoretical framework underpinning the study is mentioned. Please describe this- beyond experience in practice from the authors work, what theory or literature has shaped or influenced the design of the study

Use of passive voice

• Passive voice is used throughout, even in sentences that talk about the first author as a person. Therefore, please adjust throughout the paper to own the work you do and use active voice. Some examples you could change are “key findings are presented”

• RECRUITMENT: “were used” – change to direct tense. You are already saying one of the authors was involved in recruitment so please reduce use of indirect voice/language. Also, “were emailed by”

• Data analysis: “was adopted” – can this be direct form not indirect

Results

• Row 160 “However this simply reflected” can be changed to ‘reflecting’ - however is not relevant there. Also, is this typical of the gender breakdown in the domestic violence space? if so can note that this represents the work force.

• In the table the years of work is 1-4, 5-10, 10+ = unequal split (4 years, 6 years). What about 1-5, 6-10, 10+?

• Page 11 row 201 – what does “out of the local area” mean – used in a few places but we don’t know what you are referring to.

• page 16 “being truthful” – in the quote obviously it should stay to be true to the original meaning, but in your text perhaps change the wording to explain what that means. Currently sounds like they don’t trust the client; whereas in reality it is that sometimes people feel uncomfortable, ashamed or unsafe to disclose the extent of their abuse or their situation.

• Structure of results section: Each theme title (quote) represents just part of a theme. Because it gives an incomplete picture, I recommend adding a topic sentence or paragraph to top of each theme to let the reader know what to expect.

• Theme 4: in the opening paragraph when talking about “biggest impact” can you say in what direction or the type of impact? if you say it up top the reader will know more of what is coming – easier for the reader

Discussion

• The discussion needs restructuring. Some paragraphs of the discussion only mention external literature and don’t explicitly tie in to your findings. Every paragraph in discussion should be linked to something you found and want to discuss relative to external literature – please make sure this is the case as it will strengthen your discussion. Each paragraph should have a topic sentence that tells you what the paragraph is about – linked to something you found and discussed in the results. It should also remind us that you found it – “We found” or “our study demonstrates/ higlights” – make it easy for the reader to see how what you discussed in the results shows up again in the discussion, and make it clear what the take away is – help the reader interpret your results in the context of external literature. The reader won’t always remember what you found vs. what the external literature says so be explicit about when it is your own finding vs what others have said.

• page 30 last sentence: “This study has highlighted” – why are there two citations if you are actually talking about the findings of your own study?

• In the opening paragraph of the discussion, and again in the conclusions, please remind the reader the context/location “In this UK-based study of XXX, we found that…” or something like that.

• p 32 row 709 – “despite these positives” – this seems unrelated to the men’s care content that is just before it. . could be better as start of a new paragraph

Limitations

• how might it have influenced it?

• all female workforce- worth mentioning that it is typical in that field so the sample is broadly representative of the workforce

Conclusion

• Opening sentence – specify the field – domestic violence services. and add that you found it “our study shows” for example. Otherwise it’s a generic statement.

• I think first two paragraphs of conclusion would be better placed at top of the discussion section as they would give a overarching overview of what you found.

6. PLOS authors have the option to publish the peer review history of their article (what does this mean?). If published, this will include your full peer review and any attached files.

**Do you want your identity to be public for this peer review?** For information about this choice, including consent withdrawal, please see our Privacy Policy.

Reviewer #1: **Yes: **Maria Wemrell

Reviewer #2: No

---

## [Decision Letter · Decision Letter 1]

1 Mar 2022

PGPH-D-21-00908R1

Staff perspectives on the impact of COVID 19 on the delivery of specialist domestic abuse services in the UK: A qualitative study

Dear Dr. Haighton,

Thank you for submitting your manuscript to PLOS Global Public Health. After careful consideration, we feel that it has merit but does not fully meet PLOS Global Public Health’s publication criteria as it currently stands. Therefore, we invite you to submit a revised version of the manuscript that addresses the points raised during the review process.

Thank you for submitting your revisions. We would like you to make some minor edits before we can accept this article for publication - please see the reviewer's comments and address them.

We look forward to receiving your revised manuscript.

Kind regards,

Tia M. Palermo

Academic Editor

Journal Requirements:

Additional Editor Comments (if provided):

Reviewers' comments:

Reviewer's Responses to Questions

**Comments to the Author**

1. If the authors have adequately addressed your comments raised in a previous round of review and you feel that this manuscript is now acceptable for publication, you may indicate that here to bypass the “Comments to the Author” section, enter your conflict of interest statement in the “Confidential to Editor” section, and submit your "Accept" recommendation.

Reviewer #2: (No Response)

2. Does this manuscript meet PLOS Global Public Health’s publication criteria? Is the manuscript technically sound, and do the data support the conclusions? The manuscript must describe methodologically and ethically rigorous research with conclusions that are appropriately drawn based on the data presented.

Reviewer #2: Yes

3. Has the statistical analysis been performed appropriately and rigorously?

Reviewer #2: N/A

4. Have the authors made all data underlying the findings in their manuscript fully available (please refer to the Data Availability Statement at the start of the manuscript PDF file)?

Reviewer #2: Yes

5. Is the manuscript presented in an intelligible fashion and written in standard English?

Reviewer #2: Yes

6. Review Comments to the Author

Reviewer #2: Thank you for sharing the revised draft of your paper. This version is improved and ready for publication pending a few minor revisions. Please see attached document with particular details to be addressed.

7. PLOS authors have the option to publish the peer review history of their article (what does this mean?). If published, this will include your full peer review and any attached files.

**Do you want your identity to be public for this peer review?** For information about this choice, including consent withdrawal, please see our Privacy Policy.

Reviewer #2: **Yes: **Shelly Makleff

---

## [Editor Report · Decision Letter 2]

25 Mar 2022

Staff perspectives on the impact of COVID 19 on the delivery of specialist domestic abuse services in the UK: A qualitative study

PGPH-D-21-00908R2

Dear Dr. Haighton,

We are pleased to inform you that your manuscript 'Staff perspectives on the impact of COVID 19 on the delivery of specialist domestic abuse services in the UK: A qualitative study' has been provisionally accepted for publication in PLOS Global Public Health.

Best regards,

Tia M. Palermo

Academic Editor